# The Use of the 6MWT for Rehabilitation in Children with Cerebral Palsy: A Narrative Review

**DOI:** 10.3390/jpm13010028

**Published:** 2022-12-23

**Authors:** Domenico M. Romeo, Ilaria Venezia, Margherita De Biase, Francesca Sini, Chiara Velli, Eugenio Mercuri, Claudia Brogna

**Affiliations:** 1Pediatric Neurology Unit, Fondazione Policlinico Universitario A. Gemelli, IRCCS, 00168 Rome, Italy; 2Pediatric Neurology Unit, Università Cattolica del Sacro Cuore, 00168 Rome, Italy

**Keywords:** cerebral palsy, rehabilitation, 6 min walk test

## Abstract

Assessing and improving walking abilities is considered one of the most important functional goals of physical therapy in children with cerebral palsy. However, there is still a gap in knowledge regarding the efficacy of treatment targeting the walking capacity of children with CP, as well as their responsiveness to the treatment. The 6 min walk test (6MWT) is a reliable tool to measure this function in children with CP, although less has been known about its potential efficacy to assess changes in the walking abilities associated with interventions. The aim of the present narrative review is to increase the amount of knowledge regarding the use of the 6MWT as a reliable measure to evaluate the effect of interventions on walking capacity in children with CP.

## 1. Introduction

Cerebral palsy (CP) represents a group of disorders of movement and posture caused by a nonprogressive lesion of the immature brain, which are often accompanied by impairments of sensation, cognition and communication [1]. The movement disorder has deleterious effects on speed and quality of gait. Therefore, one of the targets of therapy in children with CP is to improve walking skills and therefore activities of daily living [2].

There has been an exponential increase in intervention in children with CP, involving medical and surgical interventions, physiotherapy and occupational therapy [3,4]. Furthermore, recent studies have underlined the combination effect of robotic technologies and physical therapy on improving walking abilities as well as postural and locomotor functions in children with CP due to the child’s motivation and to the increase in the duration and intensity of programmed exercises achieved [5,6].

However, there is still a gap in knowledge regarding the efficacy of treatment targeting walking capacity in children with CP, as well as their responsiveness to the treatment [7].

The 6 min walk test (6MWT) is a reliable tool to measure walking capacity in children with CP [2,8]. It measures the maximum distance that children can quickly walk on a flat within 6 min; it was originally developed as an endurance measure in adults with chronic heart failure [9]. Recent studies have also reported on reference values of the 6MWT in both typically developing children, children with neuromuscular disorders and CP with excellent test–retest reliability [2,7,8,9,10,11]. It also provides an indirect measure of walking efficiency as functional endurance and fatigue. In the last 10 years, the 6MWT reported valuable clinical information regarding gait abilities and outcomes in children with motor impairments. Recently, Fitzgerald et al. [11] showed significant differences in walking abilities in children with CP across GMFCS levels I to III and in typically developing (TD) children by using 6MWT scores; mainly, in children with CP, those with a GMFCS level I walked on average 89 m less, and children with a GMFCS level II walked on average 142 m less, when compared with their TD peers. Furthermore, Martakis et al. [7] precisely quantified changes in the walking capacity of ambulant children with CP, considering the expected development over a period of 6 months, utilizing age-related 6MWT trajectories. These data could be used to track changes over time in response to growth and potential clinical interventions. However, the potential efficacy of the 6MWT to assess changes in the walking abilities associated with interventions has not been systematically reported.

The aim of the present narrative review is to increase the amount of knowledge regarding the use of the 6MWT as a reliable measure to evaluate the effect of interventions on walking capacity in children with CP.

## 2. Methods

### 2.1. Search Criteria

A comprehensive search of the following electronic databases was performed: MEDLINE, EMBASE, PsycINFO, CINAHL. Search terms used were “cerebral palsy”, which was combined with “6 min walk test”.

Duplicates were excluded prior to the retrieval of references. Abstracts for each reference were obtained and screened using the following criteria.

### 2.2. Inclusion Criteria

Studies were eligible for inclusion if they were written in English and human based. All the studies were first selected, looking for the presence of a clinical association between “cerebral palsy” or “CP” and “6 min walk test” or “6MWT” and reporting details of interventions. No publication date limits were set. We included studies with an age range of 0–18 years.

### 2.3. Exclusion Criteria

Studies were excluded if they were case reports or if they assessed progressive and/or neurodegenerative disease or well-defined genetic disorders.

### 2.4. Data Extraction and Analysis

The title and abstracts of the studies were independently examined for suitability by two authors (I.L., M.D.) and critically checked by a third independent reviewer (D.M.R.); conflicting viewpoints were discussed until consensus was reached.

The selected papers were further subdivided according to the type of intervention (a) studies assessing possible changes in 6MWT in CP before/after physiotherapy; (b) studies assessing possible changes in 6MWT in CP before/after medical or surgical intervention; (c) studies assessing possible changes in 6MWT in CP before/after robotics (d) studies assessing possible changes in 6MWT in CP before/after other types of intervention not classified before.

A total of 124 studies were initially identified; after a review of the full text, 87 were excluded, as they included neurodegenerative or genetic disorders (*n* = 55), as they were case reports (*n* = 15), or as they did not provide details on 6MWT (*n* = 17). The remaining 37 articles [7,12,13,14,15,16,17,18,19,20,21,22,23,24,25,26,27,28,29,30,31,32,33,34,35,36,37,38,39,40,41,42,43,44,45,46,47,48], comprising a total of 1292 children, met the inclusion criteria after a review of the full text (Figure 1).

Table 1 reports the list of the selected papers with details on the populations studied and the outcomes.

#### 2.4.1. Studies Assessing Possible Changes in 6MWT in CP before/after Physiotherapy

In this group, 12 studies were included [12,13,14,15,16,17,18,19,20,21,22,23].

Seven articles focused on “conventional” physiotherapy, trying to analyze the efficacy of intense program training, progressive resistance exercise, home training or implementation of specific muscle training [13,14,15,16,17,20,23].

In the first study [16], 37 ambulant children with spastic CP were randomized to a comparison group (*n* = 20) and an intervention group (*n* = 17) lasting 16 weeks, including stretching and progressive resistance exercise. All the children performed the 6MWT that showed significant improvements between baseline and after 16 weeks, with significant differences between the two groups.

Aviram et al. [17] studied 95 adolescents with spastic CP divided according to 2 types of rehabilitation programs: progressive resistance training program (GT) and treadmill training program (TT) performed for 6 months. The 6MWT and other functional measures significantly improved from baseline to the end of the treatment for both GT and TT groups, but the GT group showed an advantage in percentage changes.

Bleyenheuft et al. [14] assessed the effect of Hand and Arm Bimanual Intensive Therapy Including Lower Extremity (HABIT-ILE) in 24 children with unilateral spastic CP. The 6MWT reported a significant improvement between the pre- and post- HABIT-ILE assessments, while no significant differences were observed following conventional treatment.

Cohen-Holzer et al. [23] studied a population of 14 children with unilateral CP who underwent an intensive hybrid program, combining constraint therapy with bimanual training. A significant increase in 6MWT was observed, with a median increase of 81 m (*p* = 0.004) at 3 months postintervention. The authors suggested that improvement is probably associated with improved upper extremity swing during walking, which reduces the basic energy demands for walking.

Goswami and Kusumoto [15,20] evaluated the effect of home-based training. Mainly, Goswani’s interventional study assessed possible changes in 6MWT scores before and after a period of 6 months of home-centered activity-based rehabilitation in addition to an institutional physiotherapy program in 59 children aged 5–12 with spastic diplegia vs. a control group receiving conventional physiotherapy. No differences in 6MWT were observed between the two groups.

Kusumoto and colleagues [20] evaluated if loaded sit-to-stand (STS) exercises at different speeds improved walking in 16 adolescents with spastic diplegia. The patients were divided into two groups (slow-loaded STS training group and arbitrarily loaded STS training group).

The 6MWD improved after intervention in the slow-loaded STS training group (*p* < 0.05), suggesting that the decrease in speed during exercise improves the physiological cost of walking in these children.

At last, Kepenek-Varol et al. [13] compared the effect of inspiratory muscle training (IMT) combined with a conventional physiotherapy rehabilitation program (CPRP) vs. CPRP alone on pulmonary function and function abilities in 30 children with hemiplegia. After the training, pulmonary functions and 6MWT values were significantly increased in both groups, but there were no significant differences between them.

The other five studies reported on other nonconventional therapy. Two studies [12,22] used the 6MWT to verify the efficacy of treadmill training in children diagnosed with CP. Swe et al. [12] enrolled 30 children with CP to compare partial body weight supported treadmill training and over-ground training for walking ability. Both training programs produced improvements in the 6MWT. No significant differences between the two interventions were observed.

A 4-week intensive locomotor treadmill training was employed by Mattern-Baxter et al. [22] for 6 children with CP younger than 4 years of age. Three children only were able to walk for 6 min at baseline. At 1 month, all the children were able to participate in the 6MWT, with statistically significant improvements between baseline and 1-month follow-up.

A gait training program consisting of treadmill locomotion was carried out by 11 children with CP in the study by Hoffman et al. [19]. The 6MWT scores increased after training, showing an improvement in walking speed consequent to an increase in lower extremity weight after the gait training program.

Lee and colleagues [21] studied the effects of training with a sliding rehabilitation machine on the gait function of 13 children affected by spastic CP for 8-week training. Distance walked in 6MWT significantly increased after training.

The last study was conducted by Tsai et al. [18], enrolling 8 children with cerebral palsy to examine the effect of off-axis elliptical training on gait function improvement. Participants underwent a total of 18 training sessions. An increase in 6MWT distance was observed, but it was not statistically significant.

#### 2.4.2. Studies Assessing Possible Changes in 6MWT in CP before/after Medical or Surgical Intervention

Two studies reported data on the changes in 6MWT in a pediatric population of children with cerebral palsy before/after medical or surgical intervention [24,25].

Williams et al. [24] studied a population of 15 children aged 5–11 who underwent botulin toxin type A (BoNT-A) injections for spasticity management in gastrocnemius muscles. No statistically significant changes in the distance covered in the 6-MWT were observed

Grecco et al. [25] studied a population of 15 children with CP with surgical intervention to analyze the effect of 12 weeks of treadmill training on gross motor function and functional mobility. Statistically significant improvements were found in the post-training evaluation of the 6MWT regarding distance traveled.

#### 2.4.3. Studies Assessing Possible Changes in 6MWT in CP before/after Robotics

Twelve articles investigated the 6MWT scores in children with cerebral palsy before and after robotic intervention [26,27,28,29,30,31,32,33,34,35,36,37].

Five articles focused on robot-assisted gait training (RAGT) [26,27,28,29,30]. Van Hedel and colleagues [28] studied the effect of an intensive locomotor training program using RAGT with Lokomat in a population of 67 children with cerebral palsy. Patients with GMFCS level IV showed most improvements in walking-related outcomes such as WeeFim and GMFM, but no significant changes were observed on the 6MWT.

Meyer-Heim and colleagues [27] reported on the use of RAGT with Lokomat in a group of 22 children with CP. They found comparable mean improvements in distance walked (+13%) at the 6MWT, with no statistical significance between pre- and post-treatment.

Beretta et al. [29], in a retrospective study, evaluated the effect of robotic rehabilitation and physiotherapy in a group of 182 children with acquired brain injury (ABI) or CP. The 6MWT showed improvement in both ABI and CP (*p* < 0.001)

Kim et al. [26] evaluated the effect of RAGT using a wearable torque-assisted exoskeletal (Angel Legs) in three children with bilateral spastic CP. All participants showed improvement in gross motor function and increases in walking distance during the 6MWT after RAGT.

Sucuoglu et al. [30] identified that RAGT (Robogait) in combination with a conventional treatment program was significantly associated with improvements in the walking abilities of children with mild to moderate CP. Median walking endurance assessed with the 6MWT in children with GMFCS levels I–III significantly increased by 8.7%; meanwhile, there was no improvement in children with GMFCS IV-V.

Six studies assessed the feasibility of 6MWT in evaluating the effect of robotic training or wearable robotic devices in children with CP [32,33,34,35,36,37]. Sukal-Moulton and colleagues [35] used the 6MWT to compare the effects of a robotic ankle training program in 28 children with spastic CP using the IntelliStretch robotic device, combining passive stretching and active movement protocol of 1 ankle joint. Results showed statistical improvements in 6MWT scores, especially in children with GMFCS level I.

Another study centered on robotic ankle training was conducted by Chen and colleagues [32]. They enrolled 41 children aged 7–18 years to compare the outcomes of home-based robotic ankle training to laboratory-based rehabilitation. Both groups showed significant improvements in the 6MWT and other clinical and biomechanical outcomes (active dorsiflexion range of motion, pediatric balance scale, selective motor control assessment of the lower extremity), although the differences between the two groups were not statistically significant.

Wu et al. [37] enrolled 23 children aged 4–16 with CP in a randomized controlled study using ankle and leg robotic devices to facilitate weight shift and leg swing. Participants were randomly assigned to robotic integrated training or treadmill training only. The 6MWT scores significantly increased after robotic training, but not after treadmill-only training.

In a trial conducted by Kang and colleagues [33], a robot-driven downward pelvic pull was applied to a cohort of six children with CP and crouch gait while walking on a treadmill. The 6MWT showed a significant improvement at the end of the treatment. Conner et al. [36] assessed the effect of wearable resistance in strengthening walking capacity and mobility in six children diagnosed with hemiplegic or diplegic CP. The 6MWT scores statistically increased after 4 weeks of training sessions.

Cheng et al. [32] conducted a randomized crossover study centered on robotic intervention on 18 children with spastic diplegia or quadriplegia; they were randomly assigned to an 8-week repetitive passive knee movement intervention program or to a control group. Results demonstrated a decrease in muscle spasticity and a subsequent improvement in ambulatory function after training (6MWT).

The last article was conducted by Smania and colleagues [31] on 18 children with diplegia or quadriplegia using repetitive locomotor training with an electromechanical gait trainer versus a control group that followed conventional training. In the case group, there was a significant post-treatment improvement on the 6-MWT, which was maintained at the 1-month follow-up assessment. No significant changes in performance parameters were observed in the control group.

#### 2.4.4. Studies Assessing Possible Changes in 6MWT in CP before/after Other Types of Intervention

In this group, 11 studies were included [7,38,39,40,41,42,43,44,45,46,47].

In three studies [38,39,40], the authors hypothesized that an internet-based intervention could be clinically effective for patients with cerebral palsy or acquired brain injury. Maher and colleagues [38] studied, in a randomized case–control trial, the effectiveness of an 8-week internet-based intervention (Get Set) addressed to 41 adolescents with unilateral or bilateral CP. No statistical improvements in the 6MWT were observed. Knights et al. [39] studied a population of 8 children aged 9–18 with bilateral spastic CP who completed a 6-week home-based internet-platform exergame cycling program. Assessments were performed at baseline and two days postintervention. No significant changes in 6MWT were observed. Mitchell and colleagues [40], in a case–control study including 101 children with unilateral CP, elaborated an individualized daily program with specific physical activity games and upper-limb and visual–perceptual games. Intervention was associated with significant improvements in functional strength and the 6MWT.

In the study of Brien and colleagues [41], the authors wanted to evaluate the efficacy of an intensive short-duration virtual reality intervention proposed to four adolescents with mild CP. All participants showed statistically significant changes on the 6MWT in the interventional phase, maintained at 1-month follow-up.

Two studies [42,43] reported data on the changes in 6MWT after the use of orthosis. Kerkum et al. [43] studied the effects of rigid ankle foot orthoses (AFOs) and spring-like AFOs in 15 children with spastic CP. The study demonstrated that, compared to walking shoes only, all kinds of AFOs reduced knee flexion, energy cost and improved walking ability (6MWT). The efficacy of postural insoles in children with CP in improving gait efficiency was studied by Christovão et al. [42]; 20 children were randomly assigned to the control group using placebo insoles or to the experimental group using postural insoles. No changes were observed on the 6MWT immediately after placement of insoles and after 3 months of use; performances were similar for both groups.

Five studies [7,44,45,46,47] investigate the efficacy of whole-body vibration in children with CP. In two different articles, Cheng and colleagues [44,46], evaluated the effect of an 8-week program of WBV on 16 children aged 9.2 years diagnosed with spastic diplegia or quadriplegia with two similar protocols. Results showed improvements in walking performance and a reduction of muscle spasticity for at least three days after intervention. The Improvement on 6MWT progressively attenuated after the cease of the program. The efficacy of vibration therapy combined with action observation was assessed in a randomized interventional study conducted by Jung et al. [45]. The authors studied 14 children aged 4–12 years diagnosed with spastic CP. Participants were randomly divided into two groups: children who underwent the WBV combined with action observation (WBVAO) and a group following the same training protocols without action observation for a 4-week period. Both experimental and control groups presented significant improvements in the 6MWT, but the WBVAO group showed a more significant improvement compared to the WBV group.

Martakis et al. [7] monitored 6MWT scores to assess the effect of a 6-month rehabilitation program including whole-body vibration in 157 children diagnosed with CP. Children showed significant improvement in walking capacity (6MWT) at 6 months, but no statistically significant changes at 12 months. Combining WBV and conventional physiotherapy can significantly improve the 6MWT in children with CP during the treatment but not at distance.

In the last study, Telford and colleagues [47] investigated the effect of age and GMFCS level on improvements in walking capacity (6MWT) of 59 children aged 5–20 years, before and after a 20-week vibration therapy program. All participants significantly improved their scores after intervention, independent of age and GMFCS level. However, the results demonstrated that the percentage improvements were significantly greater for children with limited mobility.

## 3. Discussion

Reduced functional walking abilities are common in children with CP and may be due to pathological inefficiencies of the musculoskeletal system during gait, with alterations in tone, poor selective motor control and muscle weakness [11]. Decreased walking mobility is also associated with decreased participation in mobility, education and social activities [48]. One of the most important functional goals of physical therapy is therefore to increase walking speed and endurance [2]. For children with CP, encouraging physical activity, such as walking, should be an important component of physical therapy intervention and may assist with increasing activity and participation abilities and their quality of life [49]. Treatment strategies for improving the gait in children with CP are related to a combination of orthopedic surgery, drugs, physical therapy and other devices. However, specific validated tools to assess these improvements should also be recommended.

The 6MWT is a common, inexpensive, easy to administer and reliable tool to measure walking capacity in children with CP. Recently, it has been demonstrated as a psychometrically robust measure of walking and is feasible for use in clinical practice [50]. In this narrative review, we have described evidence of the use of the 6MWT in children with CP, to observe any possible changes in functional outcomes after interventions in these children. In most of the studies included in the present review, an improvement of scores on 6MWT was reported, mainly in the initial phases after rehabilitation, but this seemed to fade gradually after the cease of the intervention, probably because the 6MWT might require longer intervention than provided in most of the studies to obtain longer-lasting improvement.

The 6MWT has been used as an outcome measure in several types of interventions in children with CP. As expected, walking abilities measured with the 6MWT improved after 12–24 weeks of conventional physiotherapy, especially using intense program training, progressive resistance exercise, intense home-based training or implementation of specific muscle training [12,13,14,15,16,17,18,19,20,21,22,23]. The 6MWT also improved after treadmill training, both partial body weight supported and over ground, for 4 to 8 weeks, especially in younger children who probably are more likely to show improvements after training. Regular or intensive rehabilitation activities performed by children with CP are reported to lead to modification in the representation of the motor cortex with corresponding motor improvement [51]. Mainly, during the execution of active or passive movements, children with CP show a different functional cortex activation pattern in comparison with healthy children, involving the activation of additional areas of the cortex, such as the somatosensory cortex and cerebellum. The activation of these areas could justify the improvement in walking abilities followed by physiotherapy. Furthermore, pulmonary functions and the improved upper extremity swing during walking, reducing the basic energy demands for walking, could also be responsible for the improvements in walking endurance [12,23]. Therefore, the 6MWT seems to be a good tool to assess these improvements.

The 6MWT has been used in few papers only before/after medical or surgical intervention. One study [24] only reported on the use of botulin toxin, with no changes on 6MWT after 5 weeks from the injection. A further single study [25] used the 6MWT to assess an improvement in walking after orthopedic surgery (not considering the type of surgery) and 12 weeks of treadmill training; however, in this study, it was difficult to realize if the improvement observed at 6MWT was related to the surgery per se or due to the physical rehabilitation program as previously reported. The low number of children included, as well as the relatively short outcome, did not allow drawing any useful conclusion on the use of the 6MWT as a method to assess the change in walking abilities after these interventions.

On the other hand, a lot of studies used the 6MWT before and after robotic intervention to detect any possible clinical changes from baseline [26,27,28,29,30,31,32,33,34,35,36,37]. It was applied in children using RAGT and conventional physiotherapy, but nonunivocal consideration could be made, as both improvement (especially in children with GMFCS level II–III) and stability of 6MWT parameters were reported at the end of the studies. The different protocols of RAGT in terms of the session time and the length of the program could have influenced the potential beneficial effect. The 6MWT was also used during other specific treatments (ankle–leg–pelvic robotic devices or gait trainer), and an improvement in the distance covered, such as other motor function outcomes, was observed regardless of the type of device or the length of the program. These devices, combining passive stretching and active movement of specific joints, improved muscle coordination, posture and stability during walking in children with CP, as reported by the improvements of the 6mWT scores.

The 6MWT was used in other types of intervention, such as games and virtual reality activities, orthoses and whole-body vibration [7,38,39,40,41,42,43,44,45,46,47]. The use of internet-based intervention or intensive short-duration virtual reality intervention for 4 to 20 weeks reported significant improvements in the 6MWT when the children were involved in gross motor activity games. Short, intense and repeated activities may have increased cardiovascular fitness, which could contribute to improved functional strength and walking endurance [40]. Repetition has been shown to be an important aspect of an exercise, and it is crucial in improving performance; the repetitive practice provided by these activities thus allowed the nervous system to build on previous attempts and coordinate new muscular synergies to accomplish the task goal [41]. Furthermore, enriched environments are known to promote neuronal plasticity and offer potential for long-term learning as evidenced by adaptive cerebral plasticity consistent with significant functional motor improvements in children with CP [52]

Orthoses are commonly used in clinical practice in ambulant children with CP to improve their walking abilities. The use of different types of AFOs but not postural insoles, reducing the knee flexion angle and internal knee flexor moment during stance, improved gait efficiency and walking energy cost. The normalization of knee kinematics and kinetics maximizes gait efficiency, as observed in the improvements of the 6MWT results [43].

The efficacy of walking abilities of WBV in children with CP has been reported in few studies, reporting in general improvements in 6MWT after a period of at least 4 weeks of intervention, especially when combined with action observation or conventional therapy [45,46]. The exposure of the whole body to low-frequency, low-amplitude mechanical stimuli via a vibrating platform improves spasticity and muscle strength. The vibration stimuli simultaneously reduce activity in antagonistic muscles via reciprocal inhibition and supraspinal inhibition. WBV can also induce presynaptic inhibition, reducing the release of neurotransmitters to the motoneurons [7], thereby decreasing monosynaptic reflex excitability. In addition, by enhancing the cortical excitability of the vibrated muscle, WBV increases joint range of motion (ROM). A reduction in spasticity and the improvement of ROM might lead to improvements in motor function and walking ability measured with the 6MWT. As WBV is a form of muscle and proprioceptive training independent of the motivation of the participant [53], the increased walking abilities should be considered as a real measure of improvement.

Although the 6MWT is historically used to measure the maximum distance that children can walk in 6 min, in children with CP, it proved to be a tool for assessing changes in motor function and walking efficiency, including functional endurance and fatigue, posture and stability, after most of the different types of intervention commonly used in this population of infants. It is not possible to draw firm conclusions regarding which children with CP could benefit more than others from the use of the 6MWT due to limited evidence and heterogeneity of the included studies. Mainly, although the 6MWT is in general proposed as a functional measure for children with independent walking abilities, that is, children at GMFCS levels I–II, in most of the studies, the range of GMFCS levels was wider, with important improvements in walking abilities even in children at GMFCS level III–IV. Furthermore, a recent study showed that those children classified at GMFCS levels III–IV have their peak motor function (including walking abilities) in childhood at 7–8 years with a following functional decline, whereas individuals at levels I or II have a stable gross motor function at the same ages [54]; therefore, it should be important to follow with the 6MWT even those children with moderate CP to observe possible changes due to exercise, physical activity and participation, which are crucial to maintaining function.

The age of children may also influence the improvements in 6MWT after intervention. However, in the present review, no significant correlation between changes in 6MWT and age of children was reported. It is probable that children younger than 5 years would have reported better scores due to intervention owing to their neuroplasticity; however, few studies only included children at this age, as in most cases, children with CP achieve independent walking after 4–5 years [55] and because no specific validity of 6MWT in toddlers is reported.

In considering the validity of our conclusions, the potential effects of some methodological limitations should be considered that may have affected the analysis of the reviewed papers.

As the present study was not structured as a meta-analysis, each paper was not critically evaluated, with no specific and statistical combination of the results of all the studies reviewed.

In fact, the principal limitation of the present study is related to the variability in the cohorts in terms of the research design and the characteristics of the population studied (such as the wide age range, or the heterogeneous sample size), which made pooling data unrealizable. For this reason, we decided to use a simple yes/no answer dataset about 6MWT that allowed us to include all studies, irrespective of the methods or the criteria used.

Some studies reported low sample sizes that prevent making conclusions that can be generalized to the general population. Furthermore, some of the studies included are cross-sectional, and possible selection bias cannot be excluded.

Even with these limitations, the analysis of the studies helped to confirm that the 6MWT should be considered a real measure of changes in walking abilities after most of the interventions commonly proposed in children with CP from 5 years old onwards and with a GMFCS level I to IV. The presence of longitudinal developmental trajectories, along with age-specific distributions and reference percentiles [49], allows its systematic use in both clinical and research settings to develop intervention activities that could help to maintain or improve walking and/or overall physical activity levels. Further studies are needed to examine more in-depth the evaluation quality of the 6MWT.

## Figures and Tables

**Figure 1 jpm-13-00028-f001:**
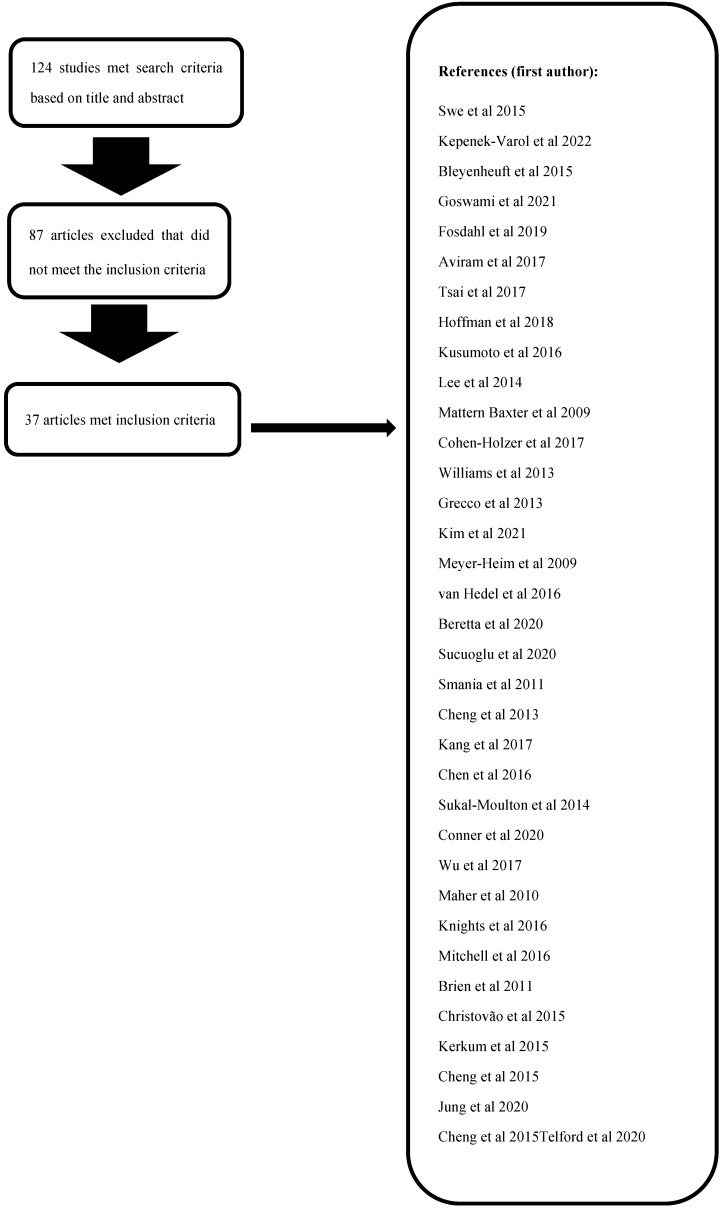
Flowchart for process of article inclusion [12,13,14,15,16,17,18,19,20,21,22,23,24,25,26,27,28,29,30,31,32,33,34,35,36,37,38,39,40,41,42,43,44,45,46,47].

**Table 1 jpm-13-00028-t001:** Study characteristics.

	Author Year	Type of Study	Sample N Male/Female	Age of Assessment Years (y) Mean ± SD	Type of CP	Type of Intervention	Outcome
1	Swe et al., 2015 [12]	Randomized controlled study	30 (20/10)	13.2 (±3.39)	CP (GMFCS II–III)	**Physiotherapy**: Partial body weight supported treadmill training/over-ground training (2-times 30 min sessions of walking training per week for 8 weeks)	Significant improvements in the 6MWT performances after a period of 4–8 weeks for both partial body weight supported treadmill training and over-ground training without significant differences between them
2	Kepenek-Varol et al., 2021 [13]	Case–control	30	7–16	Hemiplegia (GMFCS I–II)	**Physiotherapy**: Inspiratory muscle training/conventional physiotherapy versus conventional physiotherapy alone	6MWT values were significantly increased after the training in both groups No significant differences between groups
3	Bleyenheuft et al., 2014 [14]	Interventional study	24 (12/12)	6–13	Unilateral spastic cerebral palsy	**Physiotherapy**: 2 groups HABIT-ILE: immediate HABIT-ILE group (IHG, initially receiving HABIT-ILE, 10 days = 90 h), and a delayed HABIT-ILE group (DHG), which continued their conventional/ongoing treatment for an intended total duration of 90 h. Phase 2: children in the DHG were crossed over to receive HABIT-ILE, and children of the IHG were followed in their ongoing conventional therapy	Significant improvements in the 6MWT between the pre- and post-HABIT-ILE No significant differences were observed following conventional treatment
4	Goswami et al., 2021 [15]	Open-label, randomized controlled study	59 (43/16)	5–12	Spastic diplegia (GMFCS II–III)	**Physiotherapy:** Home-centered activity-based rehabilitation (walking, standing, squatting, climbing upstairs/downstairs, kicking a ball, dancing, riding a tricycle/bicycle) and institutional physiotherapy versus conventional physiotherapy	No significant differences in the 6MWT in children performing home activity compared with those who have followed a conventional physiotherapy
5	Fosdahl et al., 2019 [16]	Randomized case–control, interventional study	37 (21/16)	10.2 (±2.3)	Spastic CP (GMFCS I–III)	**Physiotherapy**: A 16-week combined stretching and progressive resistance exercise (PRE) program, followed by a 16-week maintenance program (*n* = 17) compared to traditional physiotherapy program (*n* = 20)	Significant improvements in the 6MWT between baseline and after 16 weeks No differences between the 2 groups
6	Aviram et al., 2017 [17]	Matched controlled study	95 (61/34)	14–21	Spastic CP (GMFCS II–III)	**Physiotherapy:** Circuit progressive resistance exercise training and treadmill training intervention for 30 biweekly 1 h training	Significant improvements in the 6MWT in both groups
7	Tsai et al., 2017 [18]	Interventional study	8	15.5 (± 4.1)	Not specified (GMFCS I–III)	**Physiotherapy:** Off-axis elliptical training on gait function improvement for a total of 18 training sessions (approximately 45–60 min/session) within 6 to 10 weeks (average, 8.2 ± 1.1 weeks)	No statistically significant increase in 6MWT distance
8	Hoffman et al., 2018 [19]	Exploratory study	11 (10/1)	12 ± 1	Spastic CP (10 diplegic, 1 hemiplegic)	**Physiotherapy:** Gait training program with 30 min of treadmill locomotion performed 3 days a week for 6 weeks, with a minimum of 1 day of rest between the training sessions	Significant improvements in the 6MWT after training
9	Kusumoto et al., 2016 [20]	Single-blind randomized clinical trial	16 (16/0)	12–18	Spastic diplegia (GMFCS I–III)	**Physiotherapy**: Loaded STS exercises at different speeds (slow-loaded STS training group and arbitrarily loaded STS training group); loaded sit-to-stand exercise was conducted at home for 15 min, 4 sets per day, 3–4 days per week, for 6 weeks	Significant improvements in the 6MWT in the slow-loaded STS training group
10	Lee et al., 2014 [21]	Interventional study	13 (9/4)	6–18	4 diplegia, 6 hemiplegia, 3 quadriplegia (GMFCS I–III)	**Physiotherapy**: Sliding rehabilitation machine for 8 weeks (30 min/day, 2 times/week)	Significant improvements in the 6MWT after training
11	Mattern-Baxter et al., 2009 [22]	Interventional study	6 (3/3)	2.5–3.9	3 spastic diplegia, 1 spastic quadriplegia, 1 dystonic quadriplegia, 1 hypotonic CP (GMFCS I–IV)	**Physiotherapy:** A 4-week intensive locomotor treadmill training (3 times per week for 1 h sessions consisting of 2 individualized treadmill walks)	Significant improvements in the 6MWT between baseline and 1-month follow-up
12	Cohen-Holzer et al., 2017 [23]	Interventional study	24	6–11	Unilateral cerebral palsy	**Physiotherapy**: Short-term, intensive intervention program combining constraint and bimanual training	Significant improvements in the 6MWT at post- and 3 months postintervention
13	Williams et al. 2013 [24]	Cross-sectional study	15 (10/5)	5–11 8.5 (±1.10)	Spastic diplegia (GMFCS I–II)	**Pharmacological therapy:** BoNT-A in the gastrocnemius muscle; 5 participants also received BoNT-A bilaterally to the medial hamstring muscles	No statistically significant increase in the 6MWT after 5 weeks from injections, neither in children receiving BoNT-A treatment in the gastrocnemius muscles nor in those who received BoNT-A treatment in the gastrocnemius and medial hamstring muscles
14	Grecco et al. 2013 [25]	Prospective study	15	8–15	Not specified (GMFCS II–III)	**Surgery**: Surgery intervention (Group 1: soft tissue versus Group 2: soft tissue and bone surgery) and 12 weeks of treadmill training	Significant improvements in the 6MWT in both groups on the post-training
15	Kim et al., 2021 [26]	Interventional study	3 (1/2)	9–16	Bilateral spastic cerebral palsy (GMFCS II–IV)	**Robotics:** RAGT using a wearable torque-assisted exoskeletal (Angel Legs) conducted for 17~20 sessions (60 min per session) and conventional rehabilitation program	Significant improvements in the 6MWT and in gross motor function after RAGT
16	Meyer-Heim et al., 2009 [27]	Single-case experimental	22 (13/9)	4.6–11.7 (8.6)	Bilateral spastic CP (GMFCS II–IV)	**Robotics:** RAGT with Lokomat (3–5 sessions of 45–60 min/week during a 3–5-week period)	No statistically significant increase in the 6MWT
17	Van Hedel et al., 2015 [28]	Retrospective study	67 (41/26)	3.9–19.9	Bilateral spastic CP: 57; unilateral spastic CP: 2; ataxic CP: 6; dystonic CP: 2 (GMFCS II–IV)	**Robotics:** Intensive locomotor training program using RAGT with Lokomat and conventional physiotherapy	No statistically significant increase in the 6MWT
18	Beretta et al., 2019 [29]	Retrospective study	182	4–18	110 with acquired brain injury (ABI) and 72 with cerebral palsy	**Robotics:** Combined treatment of RAGT with Lokomat and physical therapy (20 sessions + 20 sessions)	Significant improvements in the 6MWT in both ABI and CP
19	Sucuoglu et al., 2020 [30]	Cross-sectional study	38 (23/15)	7.8 (± 3.8)	Spastic CP (hemiplegic, diplegic, tetraplegic, *n* = 33); ataxic, *n* = 2; hypotonic, *n* = 2; mixed, *n* = 1 (GMFCS I–IV)	**Robotics:** RAGT in combination with a conventional treatment program (30 sessions of 60 min over 8–10 weeks)	Significant improvements in the 6MWT after RAGT in combination with a conventional treatment program in children with mild to moderate CP (GMFCS I–III)
20	Smania et al. 2011 [31]	Randomized controlled study	18	10–18	Diplegia or quadriplegia (GMFCS II–IV)	**Robotics**: 30 min repetitive locomotor training with an electromechanical gait trainer (gait trainer) versus 40 min of conventional physiotherapy. A total of 10 treatment sessions were performed over a 2-week period	Significant improvements in the 6MWT in gait trainer group
21	Cheng et al., 2013 [32]	Randomized, crossover study	18 (10/8)	9.5 ± 2.1	Spastic diplegia or spastic quadriplegia	**Robotics:** An 8-week repetitive passive knee movement intervention program (3 times a week) and a control group. After a 4-week wash-out period, children from 1 group were crossed over to the other group	Significant improvements in the 6MWT and decrease in muscle spasticity within 3 days postintervention
22	Kang et al., 2017 [33]	Interventional study	6 (6/0)	9–19	Not specified GMFCS Level II	**Robotics:** A robot-driven downward pelvic pull (tethered pelvic assist device (TPAD)) while walking on treadmill for 2 weeks after the last training	Significant improvements in the 6MWT
23	Chen et al., 2016 [34]	Pilot randomized comparative trial	41 (31/10)	7–18 8.7 ± 2.8	Hemiplegia: 21; diplegia: 20 (GMFCS I–III)	**Robotics:** 6-week home-based robotic ankle training with a portable rehabilitation robot versus laboratory-based rehabilitation	Significant improvements in the 6MWT in both groups
24	Sukal-Moulton et al., 2014 [35]	Retrospective study	28 (19/9)	8.2 ± 3.62	11 diplegia, 16 hemiplegia, 1 triplegia (GMFCS I–III)	**Robotics**: Robotic ankle training program (IntelliStretch robotic device) in a training program combining passive stretching and active movement protocol of 1 ankle joint 2 times per week for 75 min sessions for a total of 6 weeks compared to children previously involved in a laboratory-based intervention protocol (*n* = 12)	Significant improvements in the 6MWT in the IntelliStretch group
25	Conner et al. 2020 [36]	Interventional study	6 (5/1)	14 y 11 m	3 hemiplegic CP, 3 diplegic CP (GMFCS I–III)	**Robotics:** 10 training sessions in 4 weeks, consisting in walking on treadmill with a wearable resistance in strengthening walking capacity and mobility	Significant improvements in the 6MWT
26	Ming Wu et al., 2017 [37]	Randomized controlled trial	23 (14/9)	4–16 (10.9 ± 3.2)	Not specified (GMFCS I–IV)	**Robotics:** Ankle and leg robotic devices using a controlled force to the pelvis and legs and treadmill training versus treadmill training only	Significant improvements in the 6MWT after robotic training, but not after treadmill-only training
27	Maher et al., 2010 [38]	Randomized Controlled study	41 26/15	13.7 (±1.8)	Unilateral or bilateral cerebral palsy (GMFCS I–III)	**Others**: 8-week internet-based intervention, including exercises on social cognitive theory, incorporating education, quizzes, goal-setting, self-reflection and positive role modeling	No significant improvements in the 6MWT
28	Knights et al. 2014 [39]	Prospective	8 (6/2)	9–18	Bilateral spastic CP (GMFCS III)	**Others:** 6-week home-based internet-platform exergame cycling program (play of video games that require physical exertion)	No Significant improvements in the 6MWT
29	Mitchell et al., 2016 [40]	Case-controlled trial	101 (52/49)	8–17 (11 y 3 m ± 2 y 4 m)	Unilateral CP (GMFCS I–II)	**Others:** Individualized daily program of 30 min duration, 6 days per week for 20 weeks, with specific physical activity games (repetitive multi-joint bodyweight functional exercise) and upper-limb and visual–perceptual games versus waitlist control group	Significant improvements in the 6MWT and functional strength in experimental group compared with the control group at 20 weeks
30	Brien et al., 2011 [41]	Interventional study	4 (4/0)	16 (±2.25)	Spastic diplegia choreoathetosis (GMFCS level I)	**Others:** 1-month intensive short-duration virtual reality (VR) intervention consisting of 90 min of VR-based balance training on 5 consecutive days, in which the participants interacted with virtual objects encouraging motor abilities such as dynamic standing balance and coordination skills in walking performance, walking speed and endurance, and stair climbing	Significant improvements in the 6MWT in the interventional phase, maintained at 1-month follow-up
31	Christovão et al., 2015 [42]	Randomizedcontrolled, double-blind, clinical study	20 (10/10) Cases: 8/2 Controls: 7/3	4–12	Spastic diplegia	**Others:** Postural insoles versus placebo insoles for 3 months	No significant improvements in the 6MWT immediately after placement of insoles and after 3 months of use Performances were similar for both groups
32	Kerkum et al. 2015 [43]	Interventional study	15 (11/4)	6–14 (10 ± 2)	Spastic CP (GMFCS I–III)	**Others:** AFOs and spring-like AFOs that enhance push-off power	Significant improvements in the 6MWT, knee flexion and energy cost in all kinds of AFOs No significant differences were found between AFOs
33	Cheng et al. 2015 [44]	Crossover study	16 (8/8)	9.2 (±2.1)	Spastic diplegia or spastic quadriplegia	**Others:** 1 group received an 8-week WBV intervention (10 min/day) followed by an 8-week control condition, with a 4-week rest; the other group began the treatment sequence with the control condition to counterbalance the order effects	Significant improvements in the 6MWT, spasticity and active joint range for at least 3 days, but the effect attenuated over time after the cease of the intervention
34	Jung et al., 2020 [45]	Randomized, interventional study	14 (6/8)	4-12	Spastic CP (GMFCS I–III)	**Others:** 7 children underwent the WBV combined with action observation (WBVAO), performing 6 different actions demonstrated in the videos for a 4-week period; 7 children followed the same training protocols without action observation for a 4-week period. During the same period, all participants also performed physical therapy	Significant improvements in the 6MWT in both experimental (WBVAO) and control groups (WBV only) WBVAO group showed a more significant improvement
35	Martakis et al. 2020 [46]	Retrospective study	157 (84/73)	3–12	Not specified (GMFCS I–II)	**Others:** 6-month rehabilitation program including whole-body vibration. The assessments were performed at baseline, after 6 months of combined conventional physiotherapy and WBV program and after 12 months without training	Significant improvements in the 6MWT at 6 months, while no statistically significant changes at 12 months
36	Cheng et al. 2015 [47]	Complete crossover design	16 (9/7)	9.8 ± 2.3	Spastic diplegia or spastic quadriplegia	**Others: 20 min** WBV and control condition in a counterbalanced order on 2 separate days, 1 week apart	Significant improvements in the 6MWT in the WBV treatment
37	Telford et al. 2020 [48]	Interventional study	59 (33/26)	5–20 (13.8 ± 4.3)	Not specified (GMFCS II–IV)	**Others:** Side-alternating WBV, 9 min per day, 4 times per week for 20 weeks	Significant improvements in the 6MWT after intervention, especially for children with limited mobility (GMFCS III and IV)

Cerebral palsy, CP; 6 min walk test, 6MWT; Gross Motor Function Classification System, GMFCS; Hand and Arm Bimanual Intensive Therapy Including Lower Extremity, HABIT-ILE; sit-to-stand, STS; botulin toxin typeA, BoNT-A; robot-assisted gait training, RAGT; rigid ankle foot orthoses, AFOs; whole-body vibration, WBV.

## Data Availability

Not applicable.

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
