# Peer review of "The Use of the 6MWT for Rehabilitation in Children with Cerebral Palsy: A Narrative Review"

_jpm, 2022, doi:10.3390/jpm13010028_

Round 1
Reviewer 1 Report
Dear authors
It is a good idea to check about the qualifications of a frequently used test in the field of treatment in children with CP.
the strategies for selection are well performed.
I miss the qualification of the studies (risk of bias scoring)
Having said this, I was very surprised to see this approach as a narritive review.
I understand you can look in what studies the 6 MWT is used for evaluation.
However, when talking about evaluation quality of a test, one need to embed SEM, SDD calculation and available MIC into the search, methodology, results and discussion (COSMIN).
significant change after an intervention does not say anything if the SEM and SDD is larger than the change (see Dekkers K strength test and CP)
I think the base of this paper is not yet sufficient by only looking at significancy of change. one cannot state anything about the quality of the test, because it is not clear if the treatment was the best treatment or the variability of the cohort is too large.
the authors need to add SEM,SDD, MIC information into the search, methology of selection , results and change the discussion accordingly.
Author Response
We were very pleased that the reviewers found our paper interesting and had some comments that helped to improve the manuscript. Please find enclosed a list of the changes made according to the reviewers’ suggestions (in red).
We hope that the revised version will be suitable for publication
Reviewer n.1
Dear authors
It is a good idea to check about the qualifications of a frequently used test in the field of treatment in children with CP.
the strategies for selection are well performed. I miss the qualification of the studies (risk of bias scoring). Having said this, I was very surprised to see this approach as a narritive review.
I understand you can look in what studies the 6 MWT is used for evaluation. However, when talking about evaluation quality of a test, one need to embed SEM, SDD calculation and available MIC into the search, methodology, results and discussion (COSMIN). significant change after an intervention does not say anything if the SEM and SDD is larger than the change (see Dekkers K strength test and CP) I think the base of this paper is not yet sufficient by only looking at significancy of change. one cannot state anything about the quality of the test, because it is not clear if the treatment was the best treatment or the variability of the cohort is too large. the authors need to add SEM,SDD, MIC information into the search, methology of selection , results and change the discussion accordingly.
- Thank you for this comment. However the present paper was not supposed to be a metanalysis but just a review of the findings throughout the years; this was specified in the paper and a limitation section of the study was added; furthermore we remove the section results
Reviewer 2 Report
This is a narrative review that analyzes the current knowledge on the use of the 6 MWT for rehabilitation in children with CP. The study addresses an important topic, particularly because there is little information available regarding the efficiency of therapies intended to enhance CP children's walking abilities and their responsiveness to those therapies. I have some comments/suggestions that the authors may find helpful to improve the presentation of their review.
1. Introduction:- Further context and background information on the use of the 6MWT to assess the walking capacity of children with CP in light of the existing literature is needed.
2. Could the 6MWT be used reliably to assess walking efficiency in children with CP (not just about the distance covered during the test, but about the gait pattern)? I’m not sure about that. A brief comment on this point is worthwhile.
3. It would be really intriguing if you could provide more specific examples of the steps taken to guarantee the explicit and repeatable implementation of the test in the studies that were ultimately included in the review.
4. In the discussion, I would ask the author to establish a theoretical framework and focus or context for the research topic. I can see extensive comments on the interventions rather than the measurement method (i.e., 6MWT).
5. Narrative reviews may be evidence-based, but often do not meet important criteria to help mitigate bias – frequently they lack explicit criteria for article selection, and frequently there is no evaluation of selected articles for validity. This should e addressed in the study limitations.
6. By completing the article, I am not convinced that the review is going to give me any new information to influence my decision-making concerning the use of the 6MWT in clinical or research work. Additionally, despite the paper's stated goal, I am not certain that the evaluation backs the theoretical justification for taking that test into consideration. The authors might want to consider additional aspects in their review.
Author Response
We were very pleased that the reviewers found our paper interesting and had some comments that helped to improve the manuscript. Please find enclosed a list of the changes made according to the reviewers’ suggestions (in red).
We hope that the revised version will be suitable for publication
Reviewer n. 2
This is a narrative review that analyzes the current knowledge on the use of the 6 MWT for rehabilitation in children with CP. The study addresses an important topic, particularly because there is little information available regarding the efficiency of therapies intended to enhance CP children's walking abilities and their responsiveness to those therapies. I have some comments/suggestions that the authors may find helpful to improve the presentation of their review.
- Introduction:- Further context and background information on the use of the 6MWT to assess the walking capacity of children with CP in light of the existing literature is needed.
- This has been amended
- Could the 6MWT be used reliably to assess walking efficiency in children with CP (not just about the distance covered during the test, but about the gait pattern)? I’m not sure about that. A brief comment on this point is worthwhile.
- Thank you for this interesting comment. We added a paragraph in the discussion
- It would be really intriguing if you could provide more specific examples of the steps taken to guarantee the explicit and repeatable implementation of the test in the studies that were ultimately included in the review.
- We agree with the reviewer about this issue. However due to the high number of papers included in the present review we had to minimize the data of the single studies reported in the text and in the table.
- In the discussion, I would ask the author to establish a theoretical framework and focus or context for the research topic. I can see extensive comments on the interventions rather than the measurement method (i.e., 6MWT).
- this has been amended
- Narrative reviews may be evidence-based, but often do not meet important criteria to help mitigate bias – frequently they lack explicit criteria for article selection, and frequently there is no evaluation of selected articles for validity. This should e addressed in the study limitations.
- A. A paragraph including the methodological limitations has been added in the discussion
- By completing the article, I am not convinced that the review is going to give me any new information to influence my decision-making concerning the use of the 6MWT in clinical or research work. Additionally, despite the paper's stated goal, I am not certain that the evaluation backs the theoretical justification for taking that test into consideration. The authors might want to consider additional aspects in their review.
- A. the discussion has been implemented according to the reviewers’ suggestion
Reviewer 3 Report
This narrative review offers a very interesting study on the use of the 6MWT as a reliably measure to evaluate the effect of interventions on the walking capacity in children with CP.
This narrative review is really good. Only some additional information seems useful to me:
Introduction part:
It seems interesting to me that the authors give some examples of physiotherapy interventions, knowing that they mention the Lokomat in the results.
Example of additional references:
- Maghini et al. (2014) Robotic gait training in children affected with Cerebral Palsy: Effects on motor function, gait pattern and posture.
- Drużbicki et al. (2013) Functional effects of robotic-assisted locomotor treadmill therapy in children with cerebral palsy.
- Wallard et al. (2017) Robotic-assisted gait training improves walking abilities in diplegic children with cerebral palsy.
- Wallard et al. (2018) Effect of robotic-assisted gait rehabilitation on dynamic equilibrium control in the gait of children with cerebral palsy.
Method part:
It seems important to me to specify the period of research of the articles, namely from X to X year.
For the search criteria, did you use acronyms such as CP or 6MWT? If yes, specify it in your method.
Author Response
Reviewer n. 3
This narrative review offers a very interesting study on the use of the 6MWT as a reliably measure to evaluate the effect of interventions on the walking capacity in children with CP.
This narrative review is really good. Only some additional information seems useful to me:
- Thank you for this comment
Introduction part:
It seems interesting to me that the authors give some examples of physiotherapy interventions, knowing that they mention the Lokomat in the results.
Example of additional references:
- Maghini et al. (2014) Robotic gait training in children affected with Cerebral Palsy: Effects on motor function, gait pattern and posture.
- Drużbicki et al. (2013) Functional effects of robotic-assisted locomotor treadmill therapy in children with cerebral palsy.
- Wallard et al. (2017) Robotic-assisted gait training improves walking abilities in diplegic children with cerebral palsy.
- Wallard et al. (2018) Effect of robotic-assisted gait rehabilitation on dynamic equilibrium control in the gait of children with cerebral palsy.
- thank you for this comment. A paragraph with two references was added in the introduction
Method part:
It seems important to me to specify the period of research of the articles, namely from X to X year.
- No publication date limits were set. This has been reported in the methods
For the search criteria, did you use acronyms such as CP or 6MWT? If yes, specify it in your method.
- Thank you. This has been added
Round 2
Reviewer 1 Report
Dear authors
I see the changes you did in the paper and the review becomes better structured and readable.
In a narrative review it still can be needed to label the risk of bias to interpret the studies more in depth.
You removed the results about the changes, however in the description and discussion you mention still significant changes.
You really need to put in the review all information about the psychometric soundness of the 6 MWT and can described if the 6MWT can be used as an evaluation test.
Without this the paper is adding no new information and is not opening any discussion (COSMIN guidelines are recommended)
Author Response
Dear authors
I see the changes you did in the paper and the review becomes better structured and readable.
In a narrative review it still can be needed to label the risk of bias to interpret the studies more in depth. You removed the results about the changes, however in the description and discussion you mention still significant changes.
You really need to put in the review all information about the psychometric soundness of the 6 MWT and can described if the 6MWT can be used as an evaluation test.
Without this the paper is adding no new information and is not opening any discussion (COSMIN guidelines are recommended)
- Thank you for this comment. However we wish to underline that risk of bias assessment is a key defining feature of quantitative systematic reviews that is often absent from traditional narrative reviews (See Frampton et al. Principles and framework for assessing the risk of bias for studies included in comparative quantitative environmental systematic reviews. Environmental Evidence 2022). The psychometric propriety and clinical utility of the 6MWT has been previously demonstrated (see Tyson et al 2009; added in the text and in the reference) and it was not an aim of the present study. We understand the doubts of the reviewer about the limitations of the present study in this form of narrative review, however we think that even from this kind of studies, interesting information could be deduced
Reviewer 2 Report
After the authors addressed the issues and suggestions made in the previous round, the paper significantly improved.
The corrected version did have several grammatical and syntax problems, though. Before publishing, I advise editing the paper for grammatical, syntactic, and sentence structure to adhere to the journal's standards.
Author Response
significantly improved.
The corrected version did have several grammatical and syntax problems, though. Before publishing, I advise editing the paper for grammatical, syntactic, and sentence structure to adhere to the journal's standards.
A. Thank you for this comment. A review of the English has been performed.